

# ESD Ideas: The stochastic climate model shows that underestimated Holocene trends and variability represent two sides of the same coin

Gerrit Lohmann[1,2]

[1]Alfred Wegener Institute, Helmholtz Centre for Polar and Marine Research, Bremerhaven, Germany
[2] University of Bremen, Bremen, Germany

**Correspondence:** Gerrit Lohmann (Gerrit.Lohmann@awi.de)

**Abstract.** Holocene sea surface temperature trends and variability are underestimated in models as compared to paleoclimate data. The idea is presented that the trends and variability are related which is elaborated in a conceptual framework of the stochastic climate model. The relation is a consequence of the fluctuation-dissipation theorem, connecting the linear response of a system to its statistical fluctuations. Consequently, the spectrum can be used to estimate the timescale-dependent climate
sensitivity. The non-normality in the propagation operator introduces enhanced long-term variability related to non-equilibrium and/or Earth system sensitivity.

Climate and Earth system models are widely used to evaluate the impact of anthropogenic emissions on future climate. The validation of these models by simulating different climate scenarios is essential to understand the sensitivity of the climate system to external forcing. The models are clearly unrivalled in their ability to simulate a broad range of large-scale phenomena
on seasonal to decadal time scales (Flato et al., 2013). However, the reliability of models to simulate climate variability on multidecadal and longer time scales requires additional evaluation. Climate records derived from paleo-environmental proxy-parameters facilitate the testing of models across these time scales.

Interglacial periods provide the means for evaluating the performance of general circulation models in representing sea surface temperature (SST) anomalies and trends (e.g. Lohmann et al., 2013). One key finding is that the models do not capture
the magnitude of the derived SSTs from marine proxy records in all climate simulations of the Holocene where the simulated SST trends systematically underestimate the marine proxy-based temperature (Alkenone) trends. It is suggested that a part of such discrepancies can be caused either by too simplistic interpretation of the proxy data and/or by underestimated regional responses in climate models. Fig. 1a shows the scatter plot of simulated and reconstructed SST trends for the mid-to-late Holocene, based on results obtained within the Paleoclimate Modelling Intercomparison Project PMIP2/3 (Braconnot et
al., 2007, 2012). Note that the orbital forcing has different signs at high and low latitudes (Berger, 1978). The slopes in Fig. 1a indicate that the response in the models is underestimated by an order of magnitude as compared to the SST reconstructions.





By using long-term multi-millenial climate model runs and paleoclimate data, a discrepancy is detected also with respect to variability (Fig. 1b)(see, Laepple and Huybers, 2014a,b). While most state-of-the-art climate models realistically simulate inter-annual variability (in this particular model the interannual variability is overeststimated), they underestimate variability on multidecadal to millennial time scales. This was revealed by a systematic comparison of climate model simulations, instrumental records and paleo-observations.

In order to reconcile both sensitivity and variability, a model is presented which takes into account the mean as well as the variability, based on Hasselmann (1976). Imagine that the temperature of the ocean is governed by

$$\frac{d\mathbf{T}}{dt} = -\lambda \mathbf{T} + \mathbf{Q_{net}} + \mathbf{f(t)}, \tag{1}$$

where the air-sea fluxes due to weather systems are represented by a white-noise process with zero average $< Q_{net} >= 0$ and $\delta$-correlated in time

$$< Q_{net}(t)Q_{net}(t+\tau) >= \delta(\tau) \quad . \tag{2}$$

The brackets $< \cdots >$ denote the ensemble mean. The function $f(t)$ is a time dependent deterministic forcing. We assume furthermore that $f(t) = c \cdot u(t)$ with $u(t)$ as unit step or the so-called Heaviside step function. Because $< Q_{net} >= 0$, the ensemble mean solution is

$$< \mathbf{T(t)} >= \mathbf{T(0)} \cdot \exp(-\lambda \mathbf{t}) + \frac{\mathbf{c}}{\lambda} \left(\mathbf{1} - \exp(-\lambda \mathbf{t})\right) \tag{3}$$

where we have $< T(0) >= T(0)$. As equilibrium response

$$\mathbf{\Delta T} = \lim_{\mathbf{t} \to \infty} < \mathbf{T(t)} >= \frac{\mathbf{c}}{\lambda} \tag{4}$$

which is also called equilibrium climate sensitivity.

The fluctuations can be characterized by the spectral Fourier component $\hat{T}_\omega = \frac{\hat{Q}_\omega}{(\lambda+i\omega)}$ with the spectrum

$$\mathbf{S}(\omega) = < \mathbf{\hat{T}\hat{T}^*} >= \frac{< \hat{\mathbf{Q}}\hat{\mathbf{Q}}^* >}{\lambda^2 + \omega^2} = \frac{1}{\lambda^2 + \omega^2} \tag{5}$$

showing that a too high value in $\lambda$ is related to a too low low-frequency variance. The model-data differences on long time scales suggest that feedback mechanisms and internal variability may not well represented in current climate models. The relation of (4) and (5) is related to the fluctuation-dissipation theorem connecting the linear response to the statistical fluctuations (Nyquist, 1928), and the relation of a too low sensitivity (Fig. 1a) and too low variability (Fig. 1b) is qualitatively detected. Recently, one focus of research was to identify feedback mechanisms in the Earth system enhancing the sensitivity (Stärz et al., 2016) or variability (Bakker et al., 2017).





If we include more components and feedbacks into the system, we can introduce higher values for the climate sensitivity, called Earth system sensitivity. In the spectral domain, we can consider a series of processes like (1) and the spectrum is the sum

$$\mathbf{S_n}(\omega) = \sum_i \frac{1}{\lambda_i^2 + \omega^2} \quad .$$  (6)

over the components in (5). However, in the fluid dynamical context we have typically a non-normal matrix with non-orthogonal eigenvectors extracting energy from the mean flow (Trefethen et al., 1993; Farrell and Ioannou, 1996; Palmer, 1999; Lohmann and Schneider, 1999) with $\mathbf{S}(\omega) > \mathbf{S_n}(\omega)$. Consider as an example a 2 dimensional system with a $2 \times 2$ matrix

$$\mathbf{A} = \begin{bmatrix} -1 & N \\ 0 & -5 \end{bmatrix}$$  (7)

replacing the scalar number $-\lambda$ in (1), and unit matrix in (2). The eigenfrequencies are independent on N (yellow and cyan vertical lines in Fig. 1c). However, $\mathbf{S}(\omega)$ can be amplyfied relative to $\mathbf{S_n}(\omega)$ (6) by orders of magnitude (Fig. 1c), affecting the low-frequency climate variability. As a logical next step, the spectral properties can be used to estimate the time-scale dependent climate and Earth system sensitivity.



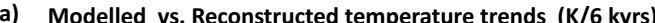

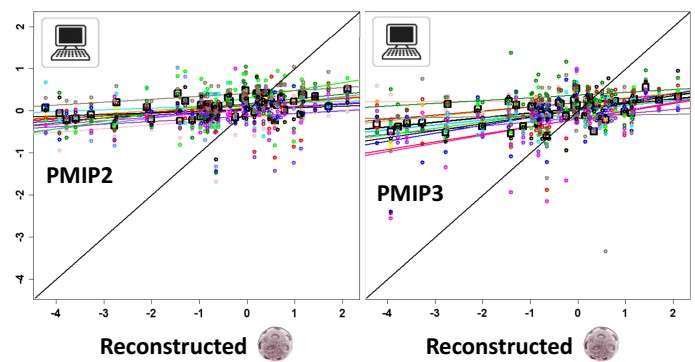

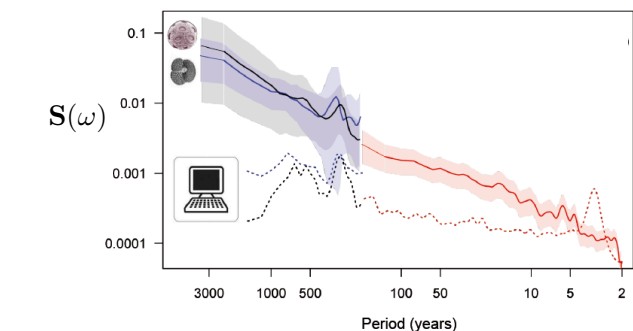

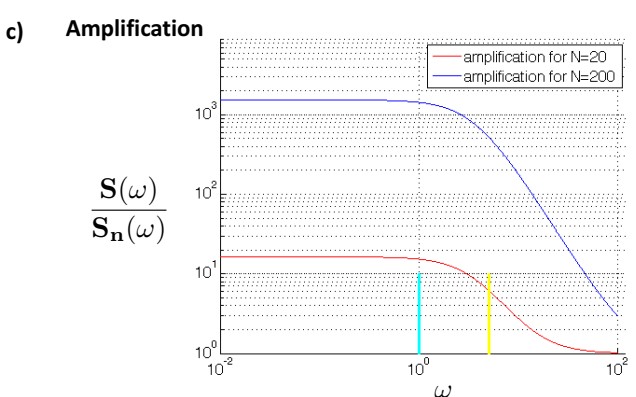

**Figure 1. a)** Global alkenone-based SST trends compared to simulated annual mean SST anomalies in the models listed in PMIP2 and PMIP3. The black squares represent the ensemble median mean and the colours correspond to a specific model (cf. Lohmann et al., 2013). Units are K per 6 kyrs. **b)** Spectrum of long-term multi-millenial climate model runs, instrumental and paleoclimate data (courtesy of T. Laepple, modified from Laepple and Huybers, 2014b). Inherent uncertainties of the paleo-observations are accounted (compare for instance the blue and black solid lines). **c)** Amplification of the spectrum $\mathbf{S}(\omega)/\mathbf{S_n}(\omega)$ of the non-normal 2 dimensional dynamics. The amplification can be several orders of magnitudes depending on the degree of non-normality related here as the parameter N in (7).



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
