# Peer review of "ESD Ideas: The stochastic climate model shows that underestimated Holocene trends and variability represent two sides of the same coin"

_Earth System Dynamics, 2018_

## Referee Comment (RC1) · Anonymous Referee #1 · 10 Aug 2018

Recommendation: I recommend that the paper not be accepted in its present form, but that it could be accepted after major revision.

Referee's General Comments This paper briefly summarizes some observational (Alkenone proxy) and modelling (GCM and ESM) results on trends and variability of global mean SST over the period 6 kyrs to present. During this period, the orbital forcing varied significantly as a function of latitude and season, but the global mean insolation remained constant. The results show that the observational SST trends are poorly defined, varying from -4K to +2K over the 6 kyr period. The modelled trends

are considerably smaller, being confined mainly to the range -1K to +1K over the same period (Figure 1(a)). The observational results also show large variability of the global mean SST with periods from 2 yr to 3000 yr, increasing towards the longer periods. It is unknown how much of this variability is externally forced and how much of it is internal. The corresponding model results show smaller variability on multidecadal to millennial periods (Figure 1(b)). In view of the poor definition of the observational trends and the lack of knowledge regarding the partitioning of the observational variability, very strong caveats should be placed on any conclusions drawn from this observational/modelling comparison. In particular, since there is no global mean orbital forcing over the 6 kyr period studied, extreme care should be exercised in drawing any conclusions from the study as to the value of climate sensitivity to greenhouse gas increase.

Referee's Specific comments

In the theoretical part of the study, a zero-dimensional stochastic model represented by Equation (1) is used in an attempt to gain conceptual understanding of the observational and modelling results described above. The term f(t) is used to describe the deterministic forcing and this is assumed to be of the form f(t) = c u(t), where c is a constant and u(t) is a unit Heaviside step function. This means that a non-zero global average forcing is assumed, in contrast to the situation prevailing in the late Holocene period 6 kyrs to present, where the global average orbital forcing is zero. From this conceptual model, it is concluded that an underestimation of variability forced by a white noise stochastic forcing implies an underestimation of climate sensitivity to the c u(t) forcing. However, this form of conceptual model does not adequately describe the climate system as forced by the late Holocene orbital forcing. A conceptual model of minimum complexity to do this would be a three-box model such as used by Stap et al. (2018) to study paleoclimate sensitivity. I recommend that such a model instead of that represented by Equation (1) be used to gain theoretical insight into the observational and modelling results described above.

Stap, L. B. et al. (2018). Modeled influence of land ice and $CO_2$ on polar amplification

and paleoclimate sensitivity during the past 5 million years. Paleoceanography and Paleoclimatology, 33. https://doi.org/10.1002/2017PA003313.

---

## Referee Comment (RC2) · M. Crucifix (Referee) · 13 Aug 2018

**1 Main appreciation**

The key message of the article is that the underestimation, by models, of low-frequency variance could be caused by a mis-representation of non-normal modes. The article describes how non-normal modes in a forced Langevin equation amplify the low-frequency variance. The conclusion of the article is that further constraints on the

time-dependent climate sensitivity matrix could be obtained from the spectrum of climate fluctuations.

The idea deserves to be formulated, although it would benefit some more critical examination of the frequency range over which it could be applied, because the underlying theory is linear and low-dimensional.

Previous investigators have indeed shown why it is not straightforward to estimate climate sensitivity by application of the fluctuation-dissipation theorem. Kirk-Davidoff (2009) provided a critique of a previous attempt by Schartz (2007) to constrain climate sensitivity from interannual variability, and Fuchs et al. (2015) as well as Cooper and Haynes (2011) provided some further technical discussion about the scope of the fluctuation-dissipation theory in atmospheric sciences. Simply put, in a simple 1-potential well system forced by Brownian motion (the Langevin equation), there is only one relaxation force in the system, which acts in a similar way at all time scales. In other words, the physical forces causing the phenomenon of relaxation (e.g.: gravitational forces in a pendulum; spring tension in a mass attached to a spring) are the same as those which determine the sensitivity to a constant forcing. This ceases to be true in the atmospheric system. Processes of relaxation at the annual time scale (which involve geophysical fluid dynamics) involve different processes than the radiative relaxation which determine climate sensitivity.

Further work on the fluctuation-dissipation theorem has since been published or shown in conferences, and would require a more systematic review, which, to be fair, is out of the scope of an "idea" paper.

Despite these reservations, it is plausible that the linear assumption expressed by the equation (1) of the article under review be indeed valid over a range of time scales greater than the interannual time scale, and hence, that the idea suggested by this author has some scope for application. However, in order to determine which range of time scales it could be, it seems necessary to provide some plausible physical interpretation of the nature of the non-diagonal terms. Indeed, the author mentions the 'fluid dynamical context', but what does it mean ?

There is also some concern about the mathematical notations. Equation (1) is originally presented with $\mathbf{T}$ as vector (if bold notation is indeed supposed to indicate a vector), with $\lambda$, a scalar. What would be the components of $\mathbf{T}$? If they are different climatic components (ocean, and atmosphere), then we need different relaxation time scales. Let us suppose that the original interpretation of equation (1) assumes $\mathbf{T}$ as a scalar, and that $\mathbf{T}$ becomes a vector only at the point of introducing equation (7). Then, we can legitimately consider that the different components of $\mathbf{T}$ correspond to different components of the climate system, in which case we would expect some non-diagonal (linear, symmetric) coupling terms. There are no such terms in matrix $\mathbf{A}$. So the reader needs to infer that the system was rotated in order to get rid of the coupling terms. What is in vector $\mathbf{Q}$ then? The second component of $\mathbf{Q}$ needs to be strictly positive, in order to excite the second component of $\mathbf{T}$, and finally generate the extra variance produced by the factor $N$. This leaves a bit too much guess work to the reader.

Assuming these questions can be answered, there is, finally, some concern about the quality and performance of spectral estimators that would be needed to do the job of estimating $\mathbf{A}$. Does the power spectrum contain enough information to constrain the non-diagonal elements of the transfer matrix? If it does, would it plausibly work given the palaeoclimate data available ?

**1.1 minor typo comments:**

use `$\langle$` and `$\rangle$` for brackets in LATEX.
**1.2 references:**

- Schwartz S. E. (2007), Heat capacity, time constant, and sensitivity of Earth's climate system, Journal of Geophysical Research, (112) doi:10.1029/2007jd008746

- Kirk-Davidoff D. B. (2009), On the diagnosis of climate sensitivity using observations of fluctuations, Atmospheric Chemistry and Physics, (9) 813–822 doi:10.5194/acp-9-813-2009

- Cooper F. C. and P. H. Haynes (2011), Climate Sensitivity via a Non-parametric Fluctuation–Dissipation Theorem, J. Atmos. Sci., (68) 937–953 doi:10.1175/2010jas3633.1

- Fuchs D., S. Sherwood and D. Hernandez (2015), An Exploration of Multivariate Fluctuation Dissipation Operators and Their Response to Sea Surface Temperature Perturbations, J. Atmos. Sci., (72) 472–486 doi:10.1175/jas-d-14-0077.1

---

## Author Comment (AC1) · 14 Aug 2018

Thanks for your detailed comments on the manuscript *ESD Ideas: The stochastic climate model shows that underestimated Holocene trends and variability represent two sides of the same coin* . In the following, I give answers to all the issues raised.

[Figure]

**Answer to the General comments:**

1. Comment: "The results show that the observational SST trends are poorly defined, varying from -4K to +2K over the 6 kyr period. The modeled trends are considerably smaller, being confined mainly to the range -1K to +1K over the same period (Figure 1(a))."

Answer: The analysis is a local one, i.e. the points at high latitudes have a general cooling trend whereas the low latitude points show a warming trend through the late Holocene. The general pattern of warming and cooling are consistent in the data and models (Figures 5a, 7a, 8a in Lohmann et al., 2013; see also Braconnot et al., 2012). In any of the analysis of the local temperature trends based on proxy reconstructions and climate simulations were taken.

Action: In the revised manuscript, I will explicitly state that it is the local temperature trend as the response to latitude-varying orbital forcing. I wrote this in the ESD manuscript at line 20: " Note that the orbital forcing has different signs at high and low latitudes (Berger, 1978)."

2. Comment: "In view of the poor definition of the observational trends and the lack of knowledge regarding the partitioning of the observational variability, very strong caveats should be placed on any conclusions drawn from this observational/modelling comparison. In particular, since there is no global mean orbital forcing over the 6 kyr period studied, extreme care should be exercised in drawing any conclusions from the study as to the value of climate sensitivity to greenhouse gas increase."

Answer: As written above, we are considering the local temperature trends based on proxy reconstructions and climate simulations. Indeed the global forcing is weak. The pattern of climate response to orbital forcing is a combination of the system's response to precession and obliquity. On the basis of the observed insolation-temperature relationship, different temperature response regimes across the Earth can be identified. Linear relationships dominate extratropical land areas whereas in midlatitude oceans, the seasonally varying mixed layer depth renders the temperature more sensitive to summer than to winter insolation (Laepple and Lohmann, 2009).

Action: In the revised manuscript, I will explicitly state that I am not analyzing the value of climate sensitivity to greenhouse gas increase.

**Answer the Referee's Specific comments:**

3. Comment: "In the theoretical part of the study, a zero-dimensional stochastic model represented by Equation (1) is used in an attempt to gain conceptual understanding of the observational and modelling results described above. The term f(t) is used to describe the deterministic forcing and this is assumed to be of the form f(t) = c u(t), where c is a constant and u(t) is a unit Heaviside step function. This means that a non-zero global average forcing is assumed, in contrast to the situation prevailing in the late Holocene period 6 kyrs to present, where the global average orbital forcing is zero. "

Answer: Indeed, the global average orbital forcing is almost zero. In the approach analyzing the climate sensitivity to external forcing such as orbital forcing, a local analysis is necessary.

Action: In the revised manuscript, I will explicitly emphasize that the global average orbital forcing is almost zero and therefore a regional analysis of the temperature trends and variability are analyzed.

4. Comment: "From this conceptual model, it is concluded that an underestimation of variability forced by a white noise stochastic forcing implies an underestimation of

climate sensitivity to the c u(t) forcing. However, this form of conceptual model does not adequately describe the climate system as forced by the late Holocene orbital forcing. A conceptual model of minimum complexity to do this would be a three-box model such as used by Stap et al. (2018) to study paleoclimate sensitivity. I recommend that such a model instead of that represented by Equation (1) be used to gain theoretical insight into the observational and modelling results described above."

Answer: I know the approach of Stap et al. (2018). Here the attempt is made to define a minimal model for global climate sensitivity and response to greenhouse gases. Here I refrain from this approach and show that the underestimated local variability in the models can be reconciled with the underestimated local responses. Future work may follow a more explicitly resolved low and high-latitude climate model (Bates, 2016; Stap et al., 2018).

Action: In the revised manuscript, I will strongly emphasize the regional aspect of the stochastic climate model in the revised version.

**References:**

Bates, J. R., 2016, Estimating climate sensitivity using two-zone energy balance models, Earth and Space Science, 3, 207-225.

Berger, A. L.: Long-term variations of daily insolation and Quaternary climatic changes, J. Atmos. Sci., 35, 2362-2367, 1978.

Braconnot, P., Harrison, S., Kageyama, M., Bartlein, P., Masson-Delmotte, V., Abe-Ouchi, A., Otto-Bliesner, B., and Zhao, Y: Evaluation of climate models using palaeoclimatic data, Nat. Clim. Change, 2, 417–424, doi:10.1038/nclimate1456, 2012.

Laepple, T., and G. Lohmann, 2009: The seasonal cycle as template for climate variability on astronomical time scales. Paleoceanography, 24, PA4201, doi:10.1029/2008PA001674

Lohmann, G., Pfeiffer, M., Laepple, T., Leduc, G., and Kim, J.-H.: A model–data comparison of the Holocene global sea surface temperature evolution, Clim. Past, 9, 1807-1839, https://doi.org/10.5194/cp-9-1807-2013, 2013.

Stap, L. B., van de Wal, R. S. W., de Boer, B., Köhler, P., Hoencamp, J. H., Lohmann, G., E. Tuenter, and L. J. Lourens, 2018: Modeled influence of land ice and CO2 on polar amplification and paleoclimate sensitivity during the past 5 million years. Paleoceanography and Paleoclimatology, 33 (4), 381-394. DOI: 10.1002/2017PA003313

---

## Author Comment (AC2) · 14 Aug 2018

Thanks for your instructive comments on the manuscript *ESD Ideas: The stochastic climate model shows that underestimated Holocene trends and variability represent two sides of the same coin* . In the following, I give answers to all the issues raised.

[Figure]

**Answers to the comments:**

1. Comment: "The key message of the article is that the underestimation, by models, of low-frequency variance could be caused by a mis-representation of non-normal modes. The article describes how non-normal modes in a forced Langevin equation amplify the low frequency variance. The conclusion of the article is that further constraints on the time-dependent climate sensitivity matrix could be obtained from the spectrum of climate fluctuations. The idea deserves to be formulated, although it would benefit some more critical examination of the frequency range over which it could be applied, because the underlying theory is linear and low-dimensional."

Answer: Indeed, the main idea behind the manuscript is to show that the underestimated local variability in the models can be reconciled with the underestimated local responses. The forced Langevin equation is the most simple dynamics to relate the spectrum with parameter $\lambda$ to the local climate response to insolation forcing, again related to the damping $\lambda$. This is related to the more general fluctuation-dissipation theorem. The fluctuation-dissipation theorem relies on the assumption that the response of a system in thermodynamic equilibrium to a small applied force is the same as its response to fluctuations. Therefore, the theorem connects the linear response relaxation of a system from a prepared non-equilibrium state to its statistical fluctuation properties in equilibrium. As compared to the termination or the Early Holocene, the linear assumption for the mid-to-late Holocene trend is a valid assumption when analyzing the SST paleoclimate data (Lohmann et al., 2013). In my approach, the missing variance in the system appears naturally from the too high $\lambda$.

The second idea is that a higher dimensional system exhibits a higher variance if the underlying dynamics is non-normal. Indeed many fluid-mechanical systems extract energy out of the mean flow and show a transient amplification Trefethen et al., 1993; Farrell and Ioannou, 1996). Without changing the eigenvalues, the system can have enhanced variance in the spectrum which is due to transient growth. Therefore, the

equilibrium climate sensitivity might be lower than the transient dynamics. The paragraph of this non-normal dynamics is admittedly sketchy, the two-dimensional dynamics is not explicitly worked out, but I found it instructive in this ESD ideas manuscript. In a simple 2-d system of the ocean thermohaline circulation (Stommel model), the system's response is far from normal introducing long-term fluctuations (Lohmann and Schneider, 2000).

You mention the frequency range over which the stochastic climate model can be applied. When looking at the spectra of the Holocene temperatures, the mid-to-late Holocene can be described very well by the linear model. Again, the termination, DO cycles or even the Early Holocene are not suitable and the variances may change over time (e.g., Wirtz et al., 2010; Wassenburg et al., 2016).

Action: In the revised manuscript, I will explicitly state that it is the local temperature trend as the response to latitude-varying orbital forcing which is quasi-linear for the mid-to-late Holocene.

2. Comment: "Previous investigators have indeed shown why it is not straightforward to estimate climate sensitivity by application of the fluctuation-dissipation theorem. Kirk-Davidoff (2009) provided a critique of a previous attempt by Schartz (2007) to constrain climate sensitivity from interannual variability, and Fuchs et al. (2015) as well as Cooper and Haynes (2011) provided some further technical discussion about the scope of the fluctuation-dissipation theory in atmospheric sciences. Simply put, in a simple 1-potential well system forced by Brownian motion (the Langevin equation), there is only one relaxation force in the system, which acts in a similar way at all time scales. In other words, the physical forces causing the phenomenon of relaxation (e.g.: gravitational forces in a pendulum; spring tension in a mass attached to a spring) are the same as those which determine the sensitivity to a constant forcing. This ceases to be true in the atmospheric system. Processes of relaxation at the annual time scale (which involve

geophysical fluid dynamics) involve different processes than the radiative relaxation which determines climate sensitivity."

Answer: Thanks a lot for these hints. Indeed, several articles are dealing with the FDT, but I have not seen a contribution to exploring the Holocene trends and variability. One motivation of the stochastic climate model by introducing $\lambda$ in the response as well as in the fluctuation is that it provides a framework for further GCM studies. Preliminary analyses of high-resolution climate models indicate a higher local SST variance as well as a more heterogeneous, enhanced SST response to external forcing.

Action: In the revised manuscript, I will explicitly mention the goal of the FDTs. As you wrote, a comprehensive overview "would require a more systematic review, which, to be fair, is out of the scope of an "idea" paper". I will mention the benefit of simple models in guiding us to analyze comprehensive models' sensitivity and variability.

3. Comment: "Despite these reservations, it is plausible that the linear assumption expressed by the equation (1) of the article under review be indeed valid over a range of time scales greater than the interannual time scale, and hence, that the idea suggested by this author has some scope for application. However, in order to determine which range of time scales it could be, it seems necessary to provide some plausible physical interpretation of the nature of the non-diagonal terms. Indeed, the author mentions the 'fluid dynamical context', but what does it mean ?"

Answer: Correct. As mentioned in comments 1. and 2., the range of timescales is given by the mid-to-late Holocene. The plausible physical interpretation of the nature of the non-diagonal terms is related to the extraction of energy from a mean state / mean flow. In terms of the simple Stommel model, this is the mean ocean circulation. In fluid dynamical context, this has been discussed in terms of shear flow instabilities (e.g., Trefethen et al., 1993).

Action: In the revised manuscript, I will explicitly mention the physics of the non-normal dynamics, but will try to reduce the number of references in this direction.

4. Comment: "There is also some concern about the mathematical notations. Equation (1) is originally presented with T as vector (if bold notation is indeed supposed to indicate a vector), with $\lambda$, a scalar. What would be the components of T? If they are different climatic components (ocean, and atmosphere), then we need different relaxation time scales. Let us suppose that the original interpretation of equation (1) assumes T as a scalar, and that T becomes a vector only at the point of introducing equation (7). Then, we can legitimately consider that the different components of T correspond to different components of the climate system, in which case we would expect some non-diagonal (linear, symmetric) coupling terms. There are no such terms in matrix A. So the reader needs to infer that the system was rotated in order to get rid of the coupling terms. What is in vector Q then? The second component of Q needs to be strictly positive, in order to excite the second component of T, and finally generate the extra variance produced by the factor N. This leaves a bit too much guess work to the reader."

Answer: Sorry. The notation in (1) was meant to be for a scalar. Indeed, the vector is only introduced with (7). The vector Q is related to the variances of the individual components. In the ESD ideas manuscript, I have not specified it explicitly and normalized it to one.

Action: In the revised manuscript, I will explicitly mention Q to avoid guesswork to the reader. Furthermore, it will be clearly stated that (1) is a scalar stochastic differential equation.

5. Comment: "Assuming these questions can be answered, there is, finally, some

concern about the quality and performance of spectral estimators that would be needed to do the job of estimating A. Does the power spectrum contain enough information to constrain the non-diagonal elements of the transfer matrix? If it does, would it plausibly work given the palaeoclimate data available ?"

Answer: In the paper, the spectra in Fig. 1c are calculated analytically. For real problems, the estimation of A can be done via the POP method (Hasselmann, 1988). Then the dynamical propagator has in general a non-normal structure. The POPs can be calculated from the paleoclimate time series, which would be a logical next step. For recent climates, there exist very nice examples in the framework of (linearized) stochastically forced dynamics (e.g., Whitaker and Sardeshmukh, 1998; Kwasniok, 2004).

Action: I will try to give a short outlook in this direction.

**References:**

Hasselmann, K., 1988: PIPs and POPs: The reduction of complex dynamical systems using principal interaction and oscillation patterns. Journal of Geophysical Research: Atmospheres https://doi.org/10.1029/JD093iD09p11015

Kwasniok, F., 2004: Empirical low-order models of barotropic flow. J. Atmos. Sci., 61, 235-245.

Wassenburg, J. A., S. Dietrich, J. Fietzke, J. Fohlmeister, K. P. Jochum, D. Scholz, D. K. Richter, A. Sabaoui, C. Spötl, G. Lohmann, M. O. Andreae, A. Immenhauser, 2016: Major reorganization of the North Atlantic Oscillation during Early Holocene deglaciation. Nature Geo, 9, 602 - 605. doi:10.1038/ngeo2767

Whitaker, J. S., and P. D. Sardeshmukh, 1998: A linear theory of extratropical synoptic eddy statistics. J. Atmos. Sci., 55, 237-258.

Wirtz, K. W., G. Lohmann, K. Bernhardt, C. Lemmen, 2010: Mid-Holocene regional reorganization of climate variability: Analyses of proxy data in the frequency domain. Palaeogeography, Palaeoclimatology, Palaeoecology, 298 (3-4), 189-200.

doi:10.1016/j.palaeo.2010.09.019

Farrell, B.F., and Ioannou, P.J.: Generalized stability theory. Part I: Autonomous operators. J. Atmosph. Sci. **53**, 2025-2040, 1996.

Trefethen, L.N., Trefethen, A.E., Reddy, S.C., and Driscoll, T.A.: Hydrodynamic stability without eigenvalues. Science 261, 578-584, 1993.
* * *